# Noncontact Sleeping Heartrate Monitoring Method Using Continuous-Wave Doppler Radar Based on the Difference Quadratic Sum Demodulation and Search Algorithm

**DOI:** 10.3390/s22197646

**Published:** 2022-10-09

**Authors:** Xiao Chen, Xuxiang Ni

**Affiliations:** College of Optical Science and Engineering, Zhejiang University, Hangzhou 310027, China

**Keywords:** continuous-wave doppler radar, heart rate detection, nonlinear demodulation, heartbeat nearest neighbor search

## Abstract

Continuous-wave doppler radar, which has the advantages of simple structure, low cost, and low power consumption, has attracted extensive attention in the detection of human vital signs. However, while respiration and heartbeat signals are mixed in the echo phase, the amplitude difference between the two signals is so large that it becomes difficult to measure the heartrate (HR) from the interference of respiration stably and accurately. In this paper, the difference quadratic sum demodulation method is proposed. According to the mixed characteristics of respiration and heartbeat after demodulation, the heartbeat features can be extracted with the help of the easy-to-detect breathing signal; combined with the constrained nearest neighbor search algorithm, it can realize sleeping HR monitoring overnight without body movements restraint. Considering the differences in vital-sign characteristics of different individuals and the irregularity of sleep movements, 54 h of sleep data for nine nights were collected from three subjects, and then compared with ECG-based HR reference equipment. After excluding the periods of body turning over, the HR error was within 10% for more than 70% of the time. Experiments confirmed that this method, as a tool for long-term HR monitoring, can play an important role in sleeping monitoring, smart elderly care, and smart homes.

## 1. Introduction

Vital signs such as heartbeat and respiration, are important indicators of human health, through which we can make a preliminary judgment on human health, thus having broad application prospects in smart elderly care and smart homes. As a noncontact monitoring tool for HR detection, doppler radar has developed rapidly over the years [1,2]. In contrast to traditional contact devices, including electrocardiogram (ECG) detectors, pulse oximeters, and health monitoring bracelets, doppler radar does not require direct contact with the human body and can penetrate some coverings such as cloths and blankets, thereby realizing the detection without restraining human body movements, having many advantages in terms of protecting personal privacy, as well as not being affected by environmental conditions such as illumination intensity. In addition, its power consumption is small, the transmission power of which does not exceed a mobile phone, with no radiation hazard to the human body [3]. Because of these advantages, doppler radar has received more and more attention in health monitoring [4], such as fall detection for elderly [5,6], cough gesture recognition [7], and daily HR/RR monitoring in different states [8].

At present, there are extensive studies on the detection of respiratory rate (RR) using doppler radar [9,10,11]. The displacement of the human chest cavity caused by respiration is much larger than that of the heartbeat, making the detection of RR easier. On the contrary, since the heartbeat signal is disturbed by respiration and random body motions (RBMs), the current existing methods struggle to achieve robust and accurate HR detection.

Demodulation processing is a primary problem of HR detection. The continuous electromagnetic wave emitted from the radar is modulated by the vibration of the target chest cavity. The echo phase changed under the influence of doppler effect contains the information of the vital signs of the target, such as the chest displacement caused by respiration and heartbeat. However, there is a nonlinear relationship between the received signal and the chest displacement; in addition to the RR and HR that we need, the signal contains the mixing and harmonic components of the two, as well as RBMs. In terms of hardware, researchers have combined orthogonal structure, phase-locked structure [12], double-sideband transmission [13], and other radar structures for optimization and improvement. In the processing method, small-angle approximation, as a linear method which is distance-correlated, requires the chest displacement to be sufficiently small compared to the carrier wavelength. Complex signal demodulation (CSD) [14] avoids the null-point problem, but suffers from harmonic interference. Although arctangent demodulation [15] or the extended differentiate and cross-multiply algorithm (DACM) [16] can demodulate the phase completely, the DC offset needs to be obtained first, which usually necessitates the use of a center estimation method such as the gradient descent method [17] or Levenberg–Marquardt algorithm [18]. However, in practice, the I–Q constellation diagram sometimes does not show a regular arc shape, resulting in a large error in the center estimation [9]. Chord approximation with the PCA method does not need to trace and eliminate dynamic DC offsets [19], but require the radian of the arc to be between π/12 and π. Articles that combined the above demodulation processing method and spectrum peak detection required some specific conditions for HR detection, such as the human body being in a static state or the arc length of the IQ constellation diagram not being too short, and the amount of data in each case was usually small.

When a human is sleeping, heartbeat and breathing slow down, and the displacement of the chest cavity is reduced, accompanied by a variety of uncontrollable situations such as sleeping position changing and hand movements. Different sleep stages have different characteristics of heartbeat, breathing, and human movement. In the current sleep monitoring articles using doppler radar, most sleep stage analysis was performed through breathing and motion characteristics rather than heartbeat characteristics for detection difficulty [20]. While HR detection results with a time length of 60 min have been proposed [21], the movements of the subjects were restricted to a supine position for a long time, and the accuracy of HR detection was not determined.

In this paper, a nighttime sleep monitoring system for human vital signs without body movement restraints is proposed, which can detect the large motion, HR, and RR of a sleeping human. Specifically, to address the difficulty of HR detection, we propose a new method based on difference quadratic sum (DQS) demodulation paired with the constrained nearest neighbor search (NNS) algorithm for long-term HR monitoring, which does not need to calculate the DC offset and requires little computation. In terms of the demodulation method, by analyzing the composition of the echo data, when the subject is in a static state, a signal whose spectrum only contains four target peaks theoretically (i.e., the double frequencies of RR and HR, and the two mixed frequencies of RR and HR) is obtained, and then an extended formula is given when small-scale random motion exists. In the experiment, using a 24 GHz continuous-wave doppler radar with quadrature structure to collect, the signal was subjected to preliminary noise reduction and DQS demodulation processing. In terms of HR feature extraction, according to the numerical relationship between the target peaks of the demodulated signal spectrum, a constrained nearest neighbor search algorithm based on the characteristics of HR variation is proposed correspondingly. Using this method, HR detection experiments were carried out on motion-controllable cardboard and sleeping subjects. The experimental results verify the effectiveness and accuracy of this method. In the face of real sleep scenarios at night, although sleeping posture and movements of the human body are not restricted, the method can still achieve good results under such circumstances; hence, it is expected to be used for human daily HR monitoring, sleep monitoring, and smart elderly care in the future.

## 2. Methods

### 2.1. Demodulation Method

When the radar antenna emits a constant-frequency continuous wave, it contacts all objects including human targets within the angle range of the antenna. The echo that scatters through them has the same frequency as the transmitted signal, but the phase is modulated, which is then collected to the receive antenna and mixed and filtered to obtain intermediate frequency (IF) data of the in-phase (*I*) and quadrature (*Q*) channel (ignoring residual phase noise).
(1)$$\begin{array}{c} s_{I}\left(t\right)=A(t)cos[\theta +\frac{4\pi }{\lambda }x\left(t\right)]+DI,\\ s_{Q}(t)=A(t)\sin[\theta +\frac{4\pi }{\lambda }x\left(t\right)]+DQ, \end{array}$$
where *A*(*t*) represents the signal amplitude of two channel, *DI* and *DQ* denote the direct-current offsets in *I*/*Q* channel, and *λ* is the carrier wavelength. ${\theta =\theta }_{0}+\frac{4\pi d}{\lambda }$, where $\theta_{0}$ is phase variation at the reflection surface, and *d* is the distance between radar and target. x(t) represents the chest displacement caused by heartbeat and respiration, simplified by the cosine model as $x{(t)=X}_{r}cos\left({2\pi f}_{r}t\right){+X}_{h}cos\left({2\pi f}_{h}t\right)$, where ${\;X}_{r}$ and $X_{h}$ denote the maximum chest displacements caused by respiration and heartbeat, while $f_{r}$ and $f_{h}$ denote RR and HR.

Differentiating the obtained I/Q channel data separately yields
(2)$$\begin{array}{c} \frac{{ds}_{I}}{dt}=\frac{dA}{dt}cos\phi \;-\;A\frac{d\phi }{dt}\sin\phi ,\;\\ \frac{{ds}_{Q}}{dt}=\frac{dA}{dt}\sin\phi +A\frac{d\phi }{dt}cos\phi ,\; \end{array}$$
where $\phi =\theta +\frac{4\pi }{\lambda }x(t)$. Taking the sum of squares of Equation (2) yields
(3)$${\left(\frac{{ds}_{I}}{dt}\right)}^{2}+{\left(\frac{{ds}_{Q}}{dt}\right)}^{2}={\left(\frac{dA}{dt}\right)}^{2}{+A}^{2}{\left(\frac{d\phi }{dt}\right)}^{2}.$$

When the amplitude variation is small, e.g., $\frac{dA}{dt}=0$, the above formula can be simplified to
(4)$${\left(\frac{{ds}_{I}}{dt}\right)}^{2}+{\left(\frac{{ds}_{Q}}{dt}\right)}^{2}{=A}^{2}{\left(\frac{d\phi }{dt}\right)}^{2}={\left(\frac{4\pi A}{\lambda }\right)}^{2}{\left(\frac{dx}{dt}\right)}^{2}.$$

The authors of [19] arrived at a similar conclusion to Equation (4) from an IQ constellation diagram, using the PCA method to determine the sign of the square root. However, because the arc was fitted by the line, the sign judgment of the phase change near zero was inaccurate, rendering it inapplicable for sleep when the heart rate amplitude is smaller than daytime and the RBMs are greater.

Expressed differently,
(5)$$\begin{array}{c} {\left(\frac{dx\left(t\right)}{dt}\right)}^{2}={\left[{2\pi f}_{r}X_{r}\sin\left({2\pi f}_{r}t\right){+2\pi f}_{h}X_{h}\sin\left({2\pi f}_{h}t\right)\right]}^{2}\\ {=a\;-\;2\pi }^{2}f_{r}^{2}X_{r}^{2}\cos\left({4\pi f}_{r}t\right){-\;2\pi }^{2}f_{h}^{2}X_{h}^{2}\cos\left({4\pi f}_{h}t\right)+\\ {4\pi }^{2}f_{r}f_{h}X_{r}X_{h}\sin\left({2\pi f}_{r}t\right)\sin\left({2\pi f}_{h}t\right), \end{array}$$
where *a* is constant term. It can be seen that the formula only includes doubled terms ($2f_{r}$ and $2f_{h}$) and cross terms ($f_{h}{\;\pm \;f}_{r}$) of RR and HR. This method alleviates the interference of multiple high-order harmonics of respiration for heart rate judgment in direct processing, such that we can obtain HR by extracting two peaks whose frequencies are $f_{h}{\;\pm \;f}_{r}$, and the difference is exactly twice that of RR, which can also be used as evidence.

In addition, when there are other uncontrolled random chest movements, the square term of x in Equation (4) needs to add $\frac{dm(t)}{dt}$, where *m*(*t*) is the displacement of RBMs. When the sway exists only for a moment, it is equivalent to introducing a broadband noise peak with a small amplitude into the spectrum, and when the sway exists for a long time but is slow or when the selected time window length is short enough, it can be regarded as a uniform motion in this period [22], where $\frac{dm\left(t\right)}{dt}=v$, which will theoretically increase the amplitude of the RR and HR peaks in the spectrum.

### 2.2. Heart Rate Feature Extraction and Search Algorithm

Due to the small amplitude, the HR-related peak with frequency ${2f}_{h}$ is easily overwhelmed by noise in actual detection and is, therefore, excluded from our selection. As a result, the peaks with frequency $2f_{r}$ and the two mixed frequencies $f_{h}{\;\pm \;f}_{r}$ are regarded as the target peaks. According to the numerical relationship between the peak frequencies, the mean frequency of the two mixing peaks is equal to HR, and the difference between them is equal to the double frequency of the breath (as shown in Figure 1). Therefore, the difference of the two frequencies and amplitude of peaks can be combined to determine the two mixing peaks, and then HR can be calculated. For healthy adults, the HR is usually in the range of 60–100 bpm, while the RR is 12–24 times/min [23]. Because the HR becomes lower during sleep, under comprehensive consideration, the prior condition is set to RR in the range of 0.2–0.4 Hz, and HR in the range of 0.85–1.5 Hz.

Figure 2 shows the whole process of extracting HR and RR from the received I/Q radar signal. The left branch is the extraction process of RR, while the right branch is that of HR. Next, the vital-sign feature extraction method is introduced in detail.

For RR, the first local maximum peak after the CSD processing is taken as a reference (${RR}_{ref}$ in Figure 2). According to the spectrum processed by the demodulation method in this paper, the maximum peak adjacent to the respiratory reference value (error range $\varepsilon_{r}$) in the range of 0.4–0.8 Hz is found, which is twice the RR.

For HR, the mixing frequencies of HR and RR, i.e., $f_{h}{\;\pm \;f}_{r}$, are needed. The condition that the two peaks differ by twice the RR is taken as the basis. Figure 3 shows the HR feature extraction process. Considering the target peaks of $f_{h}{\;\pm \;f}_{r}$ and occasionally the peak of $f_{h}$ caused by RSMs, as the top three peaks in the HR range, another peak is identified whose difference satisfies the condition to make a pair, and each pair of peaks is recorded as an HR candidate cell (case 1). When there is no peak that meets the condition, the current peak considered as the HR peak is recorded in the cell alone (case 2). For the i-th candidate cell extracted from the spectrum of the *j*-th time period, the average value is recorded as $f_{j,i}$, which represents the possible value of the HR, and the two frequency peaks contained in the cell are $f_{j,i,a}$ and $f_{j,i,b}$; then,
(6)$$f_{j,i}{=\mathrm{Mean}(f}_{j,i,a}\;{,\;f}_{j,i,b}).\;$$

The array composed of the mean values of each group cells is denoted as $H_{cand}$, and the array composed of the peak values contained in each group cells is denoted as $H_{candf}$.

Usually, determining a single target peak uses peak detection, i.e., by finding the maximum peak within the range of HR. However, there are dual target peaks after DQS demodulation; thus, a parameter is needed as the selection basis. According to Equation (5), the amplitudes of the two peaks are equal theoretically; however, in practice, when the two channels have amplitude/phase imbalance or other noises, the amplitudes of the two peaks are different, and the difference can even be large. Considering that the amplitude amount of the peak, as well as the amplitude difference of the peaks apart from the first few ones, is unimportant, in order to assess the priority of peak pair selection, the priority parameter *P* is introduced, which is defined as
(7)$$P=\{\;\begin{array}{cc} {\mathrm{Rank}(A}_{j,k}) & \mathrm{Rank}\left(A_{j,k}\right)\leq 4\\ 5 & \mathrm{others} \end{array},$$
where $A_{j,k}$ is the amplitude of the *k*-th peak in *j*-th time period, and Rank(·) returns the corresponding serial number after sorting. Suppose the priority parameter of the cell is $P_{j,i}$, and the priority parameters of the two frequency peaks contained in this cell are $P_{j,i,a}$ and $P_{j,i,b}$; then, for the two cases of peak pair or a single peak in a cell, $P_{j,i}$ is defined as
(8)$$P_{j,\;i}=\{\;\begin{array}{cc} P_{j,i,a}{+P}_{j,i,b} & \mathrm{while}\;\mathrm{case}\;1\\ P_{j,i,a}+3 & \mathrm{while}\;\mathrm{case}2 \end{array}.$$

A smaller priority parameter of the cell indicates a more preferred selection and a smaller distance in the nearest neighbor search.

Next, combined with the arrays and parameters obtained above, we can use the constrained NNS algorithm outlined in Algorithm 1. On the premise that the heart rates of adjacent groups do not change much, the constraint of proximity difference is set as 0.15 Hz.

Since the first HR value is not necessarily correct and cannot be used as the starting value of the NNS search, Layer is calculated using the initialized HR vector, and the HR in the mode layer is fixed as the reference value. Here, Layer is defined to judge the difference between HR values, which is
(9)Layerj=round[(Fj-f0)/dL],
where *F* is the initialized HR vector, f0 is the base value of HR (the first value of HR used here), round(·) denotes the function that rounds to an integer, and dL is the interval between layers. Sometimes, because of exercise, the HR peak is dominant; hence, when we treat the HR peak as one of the mixing peaks, we get an HR candidate that is one breath larger than the actual HR. To distinguish this case, dL is set as 0.2 Hz in step 2 and 0.1 Hz in the loop.
**Algorithm 1**. Heartrate nearest neighbor search.Input: Hcand, Hcandf, ***P***;Output: ***H***initial heart rate array Fˆ: Fj=fj,i, while Pj,i=min(Pj,1,…,Pj,nj).calculate the Layer vector, and find the base layer value**while *H*** is empty **do**   hpos empty positions fn are replaced with values from Hcand or Hcandf according to |fn-fn-1|<εh and min(Pn)   **if** hpos has no zero value      calculate the layer vector.      **if** the number of layer ≤ 3         ***H*** = hpos, **break**      **else** set the unqualified value to zero and continue the loop.**end while**

## 3. Radar System and Experiments

In the experiment, we used an off-the-shelf commercial 24 GHz doppler radar sensor (BGT24LTR11N16, Infineon), with an ADC module (ADS1256, Texas Instruments) for analog-to-digital conversion, and STM32 (STM32F103C8T6) to control the data storage module to automatically store the data on the SD card. The data are then transferred to the computer for digital signal processing through MATLAB. Figure 4 is the device diagram and workflow block diagram of radar system. The doppler radar system first generates a stable 24 GHz continuous sine wave through the voltage-controlled oscillator (VCO), which is sent out by the transmitting antenna. The received echo passes through the low-noise amplifier and the mixer, and the intermediate-frequency signal of the two quadrature channels of I and Q can be obtained. The sampling rate is 48 Hz, the AD sampling digit is 24 bits, and the radar is powered by 5 V from the power bank.

### 3.1. Simulation Experiment of Motion-Controllable Cardboard

To verify the correctness of the theoretical inference and preliminarily determine the noise level of the system, we chose a cardboard with controllable motion as the simulation experiment target. As shown in Figure 5, the cardboard (19.4 × 12.8 cm) is fixed on the track of the precision electronically controlled translation stage, and the stepper motor controls the front and rear movement of the translation stage.

The cardboard conducts a dual-frequency sinusoidal trajectory motion ${x(t)=A}_{1}{\sin2\pi f}_{1}{t+A}_{2}{\sin2\pi f}_{2}t\;$(${\;f}_{1}\;\leq {\;f}_{2}\;$), where $A_{1}$ and $A_{2}$ are the sinusoidal motion amplitudes, and $f_{1}$ and $f_{2}$ are the motion frequencies. The chest wall displacement caused by respiration is usually in the range of 2–12 mm [24], while that caused by a heartbeat is about 0.1 mm [25]. Thus, here, we take $A_{1}$ as 2 mm, $A_{2}$ as 0.2 mm, $f_{1}$ as 0.2 Hz, and $f_{2}$ as 1.2 Hz. Figure 6a shows the performance of the obtained two orthogonal channel IF signals in the time domain. The signal is firstly filtered by a tenth-order Butterworth lowpass filter with cutoff frequency of 2 Hz, and then processed by the demodulation method mentioned in this article. Figure 6b shows the spectra of the time between 15 and 30 s. Obviously, four target peaks can be found in the maximum peaks of each frequency band, corresponding to frequencies 0.4, 1.0, 1.4, and 2.4 Hz, respectively. It is easy to calculate the two frequencies of 0.2 Hz and 1.2 Hz in the mixed motion, which is consistent with the theoretical results.

### 3.2. Unconstrained Sleep HR Monitoring

In an unconstrained and real nighttime sleep environment, low signal-to-noise ratio, uncontrolled human movements, and sleep posture change are the difficulties in noncontact HR detection. Overnight, human sleeping includes various states such as lying down, small-scale movements, and large-scale movements, among which the posture of lying down takes up most of the time. The signal strength is related to the sleeping posture. A small part of the time, there is movement interference such as small-scale movements of the hands and head or large-scale movements of the body turning over. Respiration and heartbeat are weaker and less frequent than those in the daytime; the echo signal obtained when subjects keep a side posture is weaker than that for a supine posture.

Figure 7 shows the experimental setup of sleep HR monitoring. The subject lies on the bed, and a doppler radar is fixed at about 0.8 m above the chest. The ground truth of the HR is measured using the POLAR H10 heartrate belt worn under the chest muscles, with one measured HR datum per second. POLAR H10 detects the micro-voltage signal of the heart beat through the electrode area on the elastic belt, which has good accuracy comparable to that of medical-grade ECG equipment [26]. Since it is chosen as the HR reference device, there is no need for multiple wire connections, unlike professional medical devices such as electrocardiographs. Therefore, the state of the human body during sleep is uncontrollable, and the movements are not restricted, which is closer to the real sleep situation.

Figure 8 shows the spectrum difference between exercise and lying down. Figure 8a is the spectral data of the lying down period of 60 s, containing the breathing double-frequency peak of 0.47 Hz, as well as the mixed peaks of 0.68 Hz and 1.13 Hz. Figure 8b shows the spectrum corresponding to the period that includes a large turning over action. Except for the breathing double-frequency peak of 0.45 Hz, there are no other significant target peaks. Obviously, the error of the HR characteristic found in this case is large. As the HR when turning over is of little reference, the results of this part are excluded from our algorithm. When the action amplitude is relatively small and the duration is short, it can be solved using the NNS correction algorithm in this paper.

Figure 9 shows the HR monitoring results of sleep data for about 6 h at night, including the detection results of HR (above) and RR (bottom). Since there is no RR reference device in the experiment, only the ground truth of the HR is given, which is obtained by averaging the true values in the 60 s time window. A small part of the true value of the HR is lost, which is caused by the poor contact of the heartrate belt when turning over in sleep. The part where the true value of the HR increases and decreases sharply in the picture corresponds to the turning over action. Every 60 s, data are processed as a unit. It can be seen that the results are mostly close to the true value. Of course, large errors also exist, usually when medium/large movements occur.

To show the specific situation more clearly, we focus on about 30 min of the 6 h data. The upper and middle parts of Figure 10 are the detection results of HR and large-scale movements, respectively. The value of the period when there exists a large movement is 1; otherwise, it is 0. The lower parts are the time-domain-filtered data of the I/Q channel. There are some abrupt changes in the time-domain signal (as shown in the red box in Figure 10). These obvious “faults” are caused by sleep posture change [27], which are excluded in the algorithm. In Figure 10, period 1 is in the supine state, where the signal is relatively stable. Period 2 is the side-lying state, with the signal amplitude greatly reducing. Period 3 is in the stage of a posture change from side to supine and is unstable, having frequent body movements. It can be seen that the method in this paper can still ensure stable monitoring of HR under the situation of a change in signal-to-noise ratio and the influence of intermittent turning over.

Considering the differences in breathing/heartbeat amplitude and frequency among different individuals, as well as in the sleep state of subjects on different nights, this paper measured and processed a total of nine groups of nighttime sleep data for three subjects (as shown in Table 1) using the same method, and the duration for each group of data was about 6 h. Every 60 s, data were processed as a unit with a 10 s interval. During data processing, combined with the change in signal amplitude, the periods of large movements such as turning over were removed. The HR detection results of the nine groups are shown in Table 2, providing the percentage of results for each group of relative errors as in the range of 0–10%, 10–15%, and greater than 15%. Here, relative error is expressed as
(10)$$\varepsilon =\frac{\left|{HR-HR}_{ref}\right|}{{HR}_{ref}}.\;$$

Except for group 8, the results with error within 10% in each group accounted for more than 70%. The cumulative distribution of relative error proportions is shown in Figure 11, which represents the proportion of results within a certain relative error. When the error was small, there was a large upward trend, and when the error was large, the trend slowed down, indicating mostly small errors in results. Because subject 3 maintained a side posture for a long time, resulting in an extended duration of the reduced signal, accompanied by more small and medium movements, the results were too large in some periods of time, resulting in a larger proportion of large errors.

## 4. Discussion

The above experiments show that, in a real, unconstrained sleep environment, validated by different individuals and different nights of sleep, the DQS demodulation with the constrained NNS algorithm proposed in this paper achieved good processing results.

In the moving board simulation experiment, since the moving trajectory of the cardboard was controlled, its motion amplitude and frequency were constant, representing a relatively ideal situation for our approach. It can be seen from the spectrum that the instrument and environmental noise, such as instrument quadrature channel imbalance, phase noise introduced by the circuit, and DC drift, had little impact in this case. In the human HR monitoring experiments, random body movements of the human body were the main influencing factors, especially large-scale RBMs. It can be seen at the end of Section 3.2 that the results with error within 10% accounted for more than 70%. The remaining large error results were mainly caused by RBMs. Although the time periods containing instantaneous abrupt changes (i.e., turning over) were excluded in the above results, motion detection was too simple to account for large movements with slow changes and medium-scale movements such as hand and head movements, which interfered with the results to some extent. Furthermore, there were occasionally times when the signal was weak due to sleeping with a side posture, together with being further away from the radar.

From the point of view of the formula, in real situations, people do not keep still all the time; hence, the A-related term in Equation (3) may change with time. When the RBMs are on the order of mm, the interference is reflected in that the HR peak is greater than the two mixing peaks, which will lead to deviations when judging the peak pair. Therefore, the constrained NNS algorithm was used for correction in this paper. However, for cm-level RBMs, the amplitude variation of the received signal cannot be ignored, which introduces error in the conversion from Equation (3) to Equation (4). Therefore, for the impact of RBMs, it still needs to be combined with other methods to suppress or eliminate random body motion interference, such as matching filtering [28], a dual radar system [29], or methods to weaken the signal amplitude during the motion period located by a more accurate motion detection [30], thus obtaining a better adaptive processing algorithm.

In addition, since the two quadrature channels have a certain degree of phase and amplitude imbalance, when the HR signal is relatively weak, it will interfere with the results. In the case of weak signal, the distance between radar and subjects is limited in a small range. Further research is needed to solve the problem caused by sleep posture changes, such as improving the signal-to-noise ratio in the hardware aspect.

## 5. Conclusions

In conclusion, for the difficulty of HR detection, a new noncontact sleeping HR monitoring method using continuous doppler radar based on the DQS demodulation paired with the constrained NNS algorithm was proposed. No calculation of DC offset is needed, which avoids the problem of large deviation of the center estimation when the I–Q constellation diagram presents a non-arc complex structure, or the arc length is too small. For the spectral peak characteristics after demodulation, a corresponding search algorithm is proposed. Combined with this method, a noncontact nighttime sleep monitoring system is established, which can monitor the HR, RR, and large-scale movement of individuals in a real sleep state without motion constraints. The whole method is easy to handle and can produce good detection results in unconstrained sleeping. The results with relative error within 10% account for more than 70%.

This method can realize stable sleeping vital-sign monitoring without large-scale motion interference. In the future, it can be combined with large-scale body movement processing methods, which is expected to open up broader application prospects in smart homes, smart elderly care, etc.

## Figures and Tables

**Figure 1 sensors-22-07646-f001:**
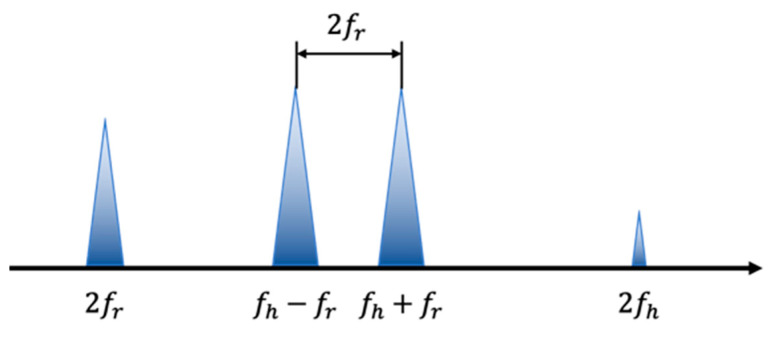
Schematic diagram of the distribution of the four peaks in theory.

**Figure 2 sensors-22-07646-f002:**
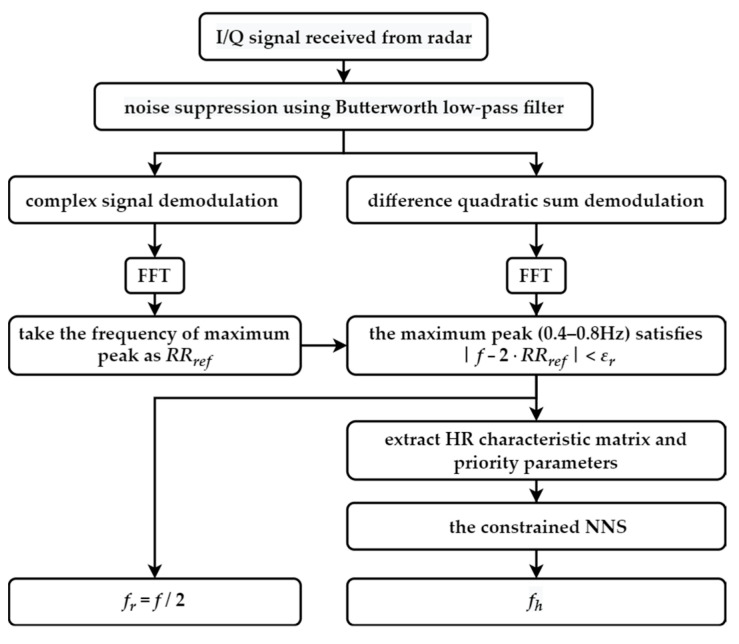
The flowchart of the whole algorithm process to extract HR and RR.

**Figure 3 sensors-22-07646-f003:**
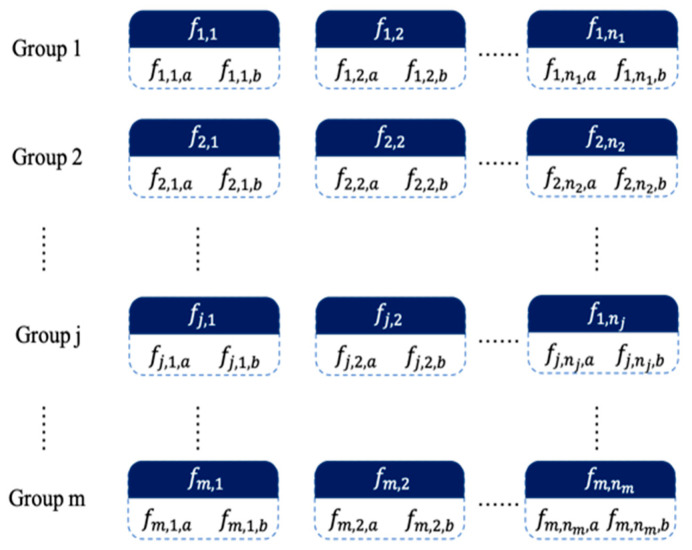
HR peak pair cells screened out from each group of spectrums. The color block above each cell is the average of cell frequencies, and the bottom is the frequency contained in the cell.

**Figure 4 sensors-22-07646-f004:**
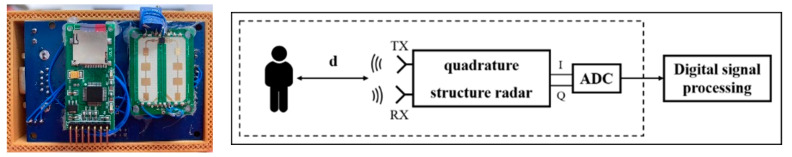
The device diagram and brief block diagram of doppler radar detection system.

**Figure 5 sensors-22-07646-f005:**
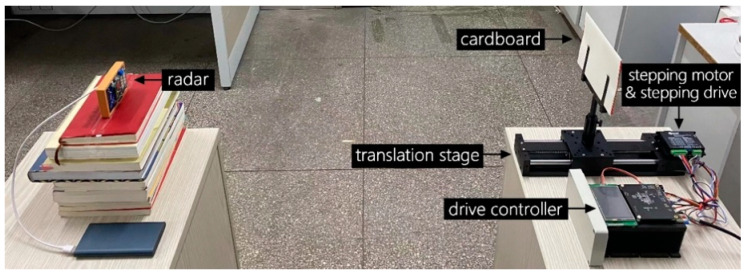
Moving cardboard simulation experiment setup.

**Figure 6 sensors-22-07646-f006:**
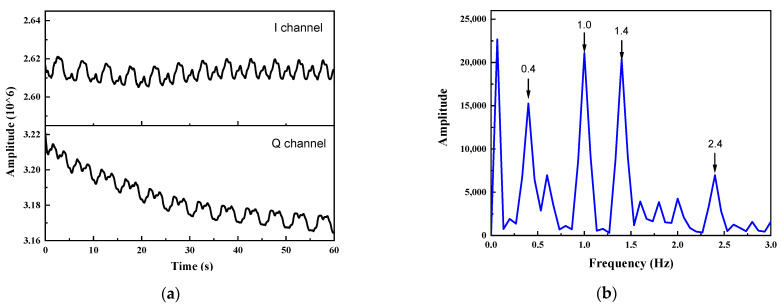
The moving board maintain a mixed sinusoidal motion with amplitude of 2 mm and 0.2 mm, and frequency of 0.2 Hz and 1.2 Hz, respectively: (**a**) the time domain diagrams of the two-channel data received by the radar; (**b**) the spectrum distribution diagrams of 15–30 s.

**Figure 7 sensors-22-07646-f007:**
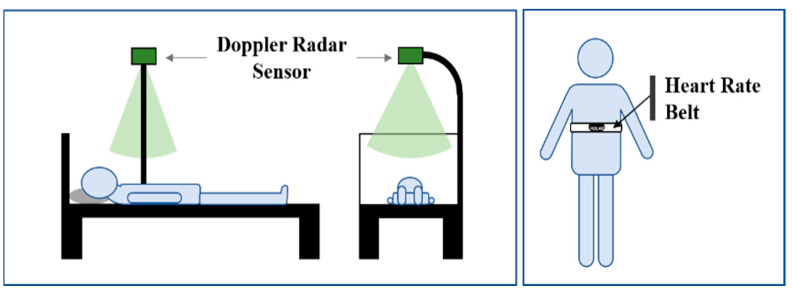
The setup for sleeping and illustration of heartrate belt.

**Figure 8 sensors-22-07646-f008:**
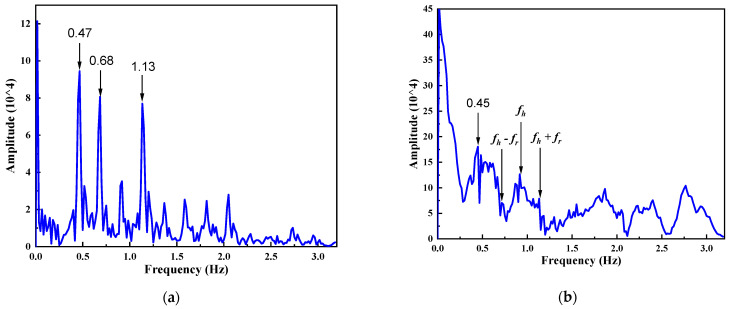
The spectrum of 60 s data in two sleeping states: (**a**) the state of lying down, where the 0.47 Hz RR-related peak and 0.68 Hz and 1.13 Hz HR-related peaks can be found; (**b**) large-scale movements state, where only the 0.45 Hz RR-related peak is obvious and HR-related peaks are drowned out by movement noise (the ground truth of HR is 56 bpm, i.e., 0.93 Hz).

**Figure 9 sensors-22-07646-f009:**
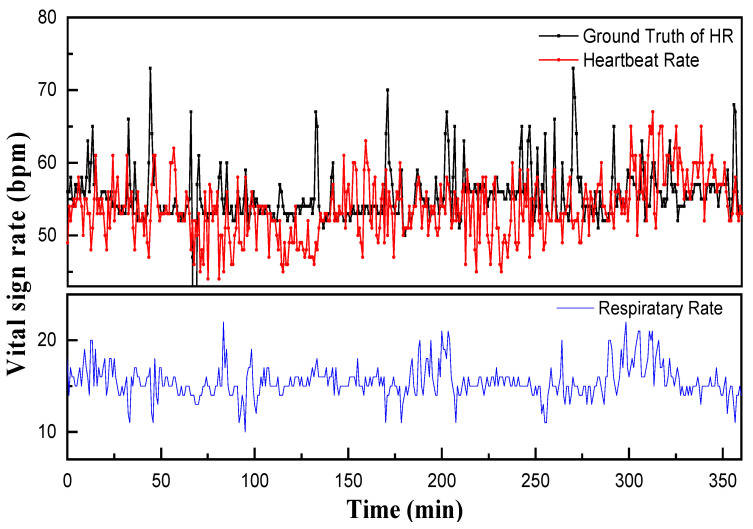
HR results in sleep state for about 6 h. The above panel depicts the HR and the true value results of HR, while the bottom panel depicts the respiration detection results.

**Figure 10 sensors-22-07646-f010:**
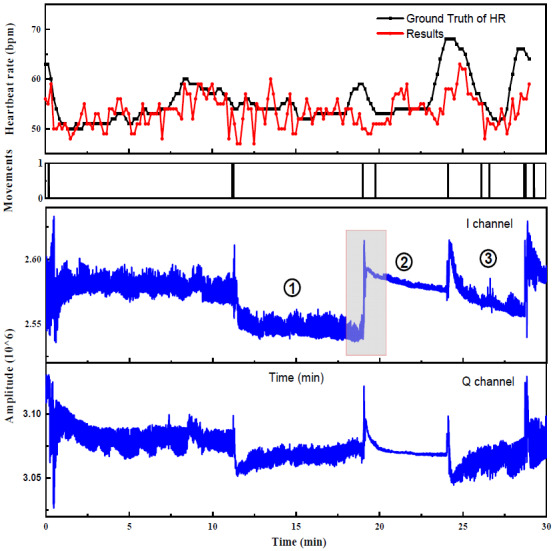
HR results in sleep state for about 30 min (**top**), motion detection results (**middle**), and time-domain diagrams of two orthogonal channels data (**bottom**). There is a corresponding relationship between the time with a large amplitude change in the time domain and the time with a large error in the obtained result. The red box shows the signal region with a posture change from supine to side, whose amplitude is greatly reduced.

**Figure 11 sensors-22-07646-f011:**
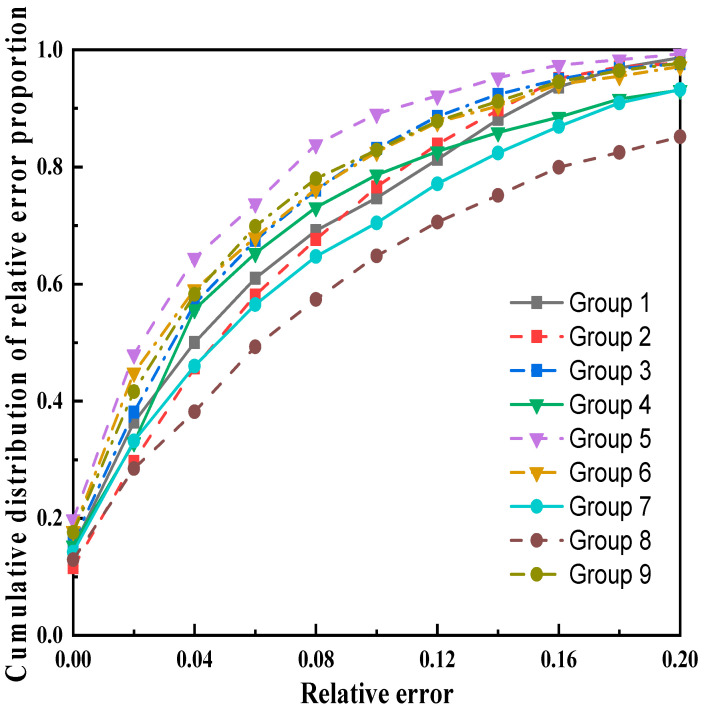
The cumulative distribution of relative error proportions for each group.

**Table 1 sensors-22-07646-t001:** The summary of the three subjects.

Subject	Gender	Age	Weight	Height	Sleep Habit
Subject 1	F	24	50	1.56	More time in a supine posture
Subject 2	M	23	74	1.74	More time in a supine posture, less movement during sleep
Subject 3	M	24	70	1.67	More time in a side-lying posture, more movement during sleep

**Table 2 sensors-22-07646-t002:** The proportions of 6 h night sleep HR detection data errors compared to the true value for the three groups.

Subject	Group	Relative Error			Mean Value of HR Reference (bpm)	$\varepsilon_{mean}$
		≤10%	10–15%	>15%		
Subject 1	1	74.7%	16.7%	8.6%	54	6.3%
	2	76.6%	16.0%	7.4%	57	6.3%
	3	83.2%	10.7%	6.1%	54	5.8%
Subject 2	4	78.7%	8.5%	12.8%	51	6.6%
	5	89.1%	7.7%	3.2%	61	4.2%
	6	82.5%	9.9%	7.6%	60	5.2%
Subject 3	7	70.4%	14.4%	15.2%	56	7.3%
	8	64.8%	12.5%	22.7%	53	9.6%
	9	82.9%	9.8%	7.3%	54	5.4%

## Data Availability

The data presented in this study are available on request from the corresponding author.

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
