# Peer review of "Noncontact Sleeping Heartrate Monitoring Method Using Continuous-Wave Doppler Radar Based on the Difference Quadratic Sum Demodulation and Search Algorithm"

_sensors, 2022, doi:10.3390/s22197646_

Round 1

Reviewer 1 Report

the work done is of very high quality with a very well described analytical study and an extensive test with a simulator and then with a set of people.

The only aspect to improve is to describe more clearly the constrained NNS algorithm that is not well understood.

An excellent work

Reviewer 2 Report

Strong points:

1. In this paper, the difference quadratic sum demodulation method is proposed.    2. According to the mixed characteristics of respiration and heartbeat after demodulation, the heartbeat features can be extracted with the help of the easy-to-detect breathing signal, and combined with the constrained nearest neighbor search algorithm, it can realize sleeping HR monitoring overnight without body movements restraint. Considering the differences in vital-sign characteristics of different individuals and the irregularity of sleep movements,    3. Data set can be appreciated: 54 hours of sleep data for 9 nights were collected from 3 subjects, which compared with ECG based HR reference equipment.    4. Accuracy is quite good After excluding the periods of body turning over, the HR error is within 10% for more than 70% of the time.   Some comments:   1. Eq (1) should be revised. DI and DQ should be put in the bracket. If they are out side the bracket, they are DC offset, not phase offset. Because (1) change, Eq.(2) to (6) should be changed, although not much changes.   2. The method in this paper has 2 stages: (1) sum of square and (2) Nearest Neighbor Search (NNS) algorithm. stage (1) is not new, some previous papers did this. Author should survey and make clear the contribution in the paper.   3. Section 2.2 should be describe by flow chart, so that reader can be easier to read.   4. Some conclusions should be make clear, why has the conclusion. Eg. line 155 and line 203.   5. Define "priority parameters" on page 5.   6. Authors should make clear for ground truth: Sampling rate? Reference? (the output of the device is HR or voltage?) Length of time window for once applying algorithms?   7. Figure 9 shows the results. However, when occuring random body movement, the accuracy is low. State-of-the-art has papers on this problem, however, authors did not mention. 70% data got accuracy of higher 90%. Is there 30% did not get high accuracy because the random body movements? Author can discuss about this.    So that, to apply this method, it requires high quality signal. In real application, author should discuss more.   8. The method can be apply for the patients having HR different from healthy subjects?

Reviewer 3 Report

The article "Noncontact sleeping heartbeat-rate monitoring method using continuous-wave doppler radar based on the difference quadratic sum demodulation and search algorithm" proposes the difference quadratic sum demodulation method  for noncontact sleeping HR monitoring. In this reviewer’s opinion, the paper needs improvements:

1- In "3. Radar system and Experiments", what mean "data accuracy is 24 bits" in the sentence "The sampling rate is 48Hz, the data accuracy is 24 bits,..."?

2- Explain the steps that you use to extract demodulated signalat frequecy from IQ received signal. I suggest that you insert a figure with workflow with all steps to extract Heart Rate and Breathing Rate starting from received IQ signal (Detail the digital signal processing of Figure 3.

3- Use of microwave and mmwave radars for health monitoring has gained importance. Insert in the Introduction examples of use Doppler Radar sensors into health monitoring: 10.3390/s17030485, 10.3390/bdcc3010003, 10.1109/JSEN.2020.3028494, 10.1109/ISC251055.2020.9239074

4- Figure 7 shows the spectrum in the region of breath rate. Insert a figure with the spectrum region of heart rate. This will help the readers understand the difference between breathing rate and heart rate.

5- Insert blankspaces between the numerical value and unit, 60s => 60 s, 0.47Hz => 0.47 Hz ...

Round 2

Reviewer 2 Report

The revised manuscript should be fine for publication.